# Study of the Interaction of a Novel Semi-Synthetic Peptide with Model Lipid Membranes

**DOI:** 10.3390/membranes10100294

**Published:** 2020-10-19

**Authors:** Lucia Sessa, Simona Concilio, Peter Walde, Tom Robinson, Petra S. Dittrich, Amalia Porta, Barbara Panunzi, Ugo Caruso, Stefano Piotto

**Affiliations:** 1Department of Pharmacy, University of Salerno, 84084 Fisciano (SA), Italy; aporta@unisa.it (A.P.); piotto@unisa.it (S.P.); 2Research Centre for Biomaterials BIONAM, University of Salerno, Via Giovanni Paolo II 132, 84084 Fisciano (SA), Italy; 3Department of Materials, ETH Zürich, 8093 Zürich, Switzerland; peter.walde@mat.ethz.ch; 4Department of Theory and Bio-Systems, Max Planck Institute of Colloids and Interfaces, D-14424 Potsdam, Germany; tom.robinson@mpikg.mpg.de; 5Department of Biosystems Science and Engineering, ETH Zurich, 4058 Basel, Switzerland; petra.dittrich@bsse.ethz.ch; 6Department of Agriculture, University of Napoli Federico II, 80055 Portici (NA), Italy; barbara.panunzi@unina.it; 7Department of Chemical Sciences, University of Napoli Federico II, 80126 Napoli, Italy; ugo.caruso@unina.it

**Keywords:** peptide, MD, GUV, LUV, azo-amino acid

## Abstract

Most linear peptides directly interact with membranes, but the mechanisms of interaction are far from being completely understood. Here, we present an investigation of the membrane interactions of a designed peptide containing a non-natural, synthetic amino acid. We selected a nonapeptide that is reported to interact with phospholipid membranes, ALYLAIRKR, abbreviated as ALY. We designed a modified peptide (azoALY) by substituting the tyrosine residue of ALY with an antimicrobial azobenzene-bearing amino acid. Both of the peptides were examined for their ability to interact with model membranes, assessing the penetration of phospholipid monolayers, and leakage across the bilayer of large unilamellar vesicles (LUVs) and giant unilamellar vesicles (GUVs). The latter was performed in a microfluidic device in order to study the kinetics of leakage of entrapped calcein from the vesicles at the single vesicle level. Both types of vesicles were prepared from a 9:1 (mol/mol) mixture of POPC (1-palmitoyl-2-oleoyl-*sn*-glycero-3-phosphocholine) and POPG (1-palmitoyl-2-oleoyl-*sn*-glycero-3-phospho(1′-*rac*-glycerol). Calcein leakage from the vesicles was more pronounced at a low concentration in the case of azoALY than for ALY. Increased vesicle membrane disturbance in the presence of azoALY was also evident from an enzymatic assay with LUVs and entrapped horseradish peroxidase. Molecular dynamics simulations of ALY and azoALY in an anionic POPC/POPG model bilayer showed that ALY peptide only interacts with the lipid head groups. In contrast, azoALY penetrates the hydrophobic core of the bilayers causing a stronger membrane perturbation as compared to ALY, in qualitative agreement with the experimental results from the leakage assays.

## 1. Introduction

Membrane interacting peptides are an exciting topic of research, because they cover different classes of peptides with several biological activities. The interactions between peptides and lipid membranes are involved in many critical biological processes [1,2]. Depending on their structural characteristics, different peptides employ different mechanisms of interaction with the membrane, causing membrane alteration or permeation [3,4]. Moreover, during their interactions, peptides and membranes may undergo a sequence of structural changes. Even for short and linear peptides, the molecular details of the process are often not completely understood. Numerous studies have been carried out to clarify the interactions of peptides with lipid bilayers, when considering the position, orientation, structure, and the effects on the surrounding lipids [5,6,7,8,9,10]. A better understanding of peptide-membrane interactions at a molecular level is not only essential in the study of various biological processes, but it could also help in designing peptides with specific functionalities that may be exploited for therapeutic applications. We performed theoretical and experimental studies using model lipid membranes to develop a novel semi-synthetic peptide with a direct effect on membrane perturbation [11]. Here, we report the results of such investigations highlighting the interactions of a designed peptide with lipid membranes. We selected a membrane interacting nonapeptide, ALYLAIRKR (abbreviated as ALY) [12], after a screening of peptides in the database YADAMP [13]. This peptide is known to interact with liposomal membranes and cause an increase in membrane permeability proportional to the peptide concentration. It is known that linear peptides and membrane proteins affect membrane structure [14,15,16,17], but little is known regarding peptides with azo-modified amino acids. 

We used this peptide as a reference peptide to develop a novel modified peptide (azoALY) replacing the tyrosine (“Y”) in the ALY sequence with an unnatural amino acid bearing an azo group, which may affect the membrane permeability and the antimicrobial activity. We chose to modify the tyrosine residue of ALY by replacing one of the hydrogen atoms of the phenyl ring in the side chain with an azobenzene group (via a diazo coupling reaction).

The novel azo-amino acid (azoTyr) was then employed in order to synthesize the peptide chain without further modifications. The choice of azobenzene in the peptide chain was determined by its hydrophobicity and intrinsic antimicrobial properties. In our previous works [18,19], we have already studied the antibacterial and antifungal activity of the azobenzene group. Therefore, a modified peptide might have antimicrobial activity due to the presence of azobenzene, and it might act as a prodrug, releasing the antimicrobial azo compound in vivo. However, this work is not focused on the antimicrobial aspects, rather on the membrane perturbation effect. With this purpose, molecular dynamics (MD) simulations are a useful method for studying peptide-membrane interactions. They provide a detailed description of the processes at a molecular level, in which all of the components can be studied, as well as being able to visualize their organization and dynamics [20]. 

In the first part of the work, we used MD simulations to understand the different modes of interactions between the two peptides (ALY and azoALY) and a model bilayer. We employed a mixture of POPC/POPG at a molar ratio of 9:1 to study the interaction of the peptides with the surface of lipid bilayers, according to previous studies [21,22]. POPC/POPG lipids are frequently used to build a model of bacterial membranes (see, e.g., [23,24,25]). It is relevant to evidence that, although POPC is widely used in both molecular dynamics simulations and vesicle preparations, it is not common to find it in bacterial membranes. Still, the preparation of POPC/POPG membranes offers two essential advantages. First, the results can be easily compared with similar settings in the scientific literature. This kind of vesicle is prepared in a standard and easy way due to low phase transition temperature of the lipids. Second, it allows for the building of a negatively charged membrane without introducing curvature stress. The MD trajectories allowed for us to analyse the dynamics of the peptides in the lipid bilayer and their effects on the lipid molecules. Membrane-active peptides can induce an immediate and complete release of entrapped solutes from the aqueous interior of lipid vesicles (liposomes) into the bulk solution by forming membrane pores [26,27]. In the second part of the work, we prepared vesicles with the same composition (POPC/POPG, 9:1) to assess the amount of membrane perturbation that is caused by the interaction with the peptides. Two different approaches were used. First, we estimated the release of entrapped calcein that is induced by the two peptides from giant unilamellar vesicles (GUVs). Secondly, we performed measurements of the leakage of entrapped horseradish peroxidase isoenzyme C enzyme (HRPC) from large unilamellar vesicles (LUVs) after peptide addition. 

## 2. Materials and Methods

### 2.1. Selection of the Reference Peptide ALY

The database YADAMP was used to select a membrane-interacting peptide [13]. The criteria for the selection were: (i) a tyrosine in the amino acid sequence; (ii) a sequence length of ≤10 residues; (iii) a helicity value higher than 4.5; and, (iv) a cell-penetrating potential value higher than 0.5.

The helicity value is estimated on a scale range between 0 and 9, whereby the value 9 corresponds to the highest probability of α-helix secondary structure formation [28]. The cell-penetrating value is estimated on a scale range between 0 and 1, whereby 1 corresponds to the highest probability of a peptide in order to penetrate a membrane and 0 indicates the impossibility to enter a membrane [29].

We selected the peptide ALYLAIRKR (H-Ala-Leu-Tyr-Leu-Ala-Ile-Arg-Lys-Arg-NH_2_) [12]. The peptide is abbreviated as ALY, whereas the modified peptide was named azoALY.

### 2.2. Molecular Dynamics (MD) Simulations

The three-dimensional structures of the peptides were built by MD simulations. In MD simulations, Newton’s equations of motion are solved at each step of the atom movement, which is probably the most reliable method for investigating protein interactions. MD simulations were performed while using the script protein_folding_by_MD.js, accessible in the Abalone software 2.1.4.2 Version [30], setting the AMBER94 force field, the temperature to 350 K, and the implicit water model (continuum solvent). After force field assigning, we performed structure optimization to avoid overlapping. The simulation time of 10 ns was enough to reach the native protein conformation. The interactions of peptides with membranes were evaluated on the model membrane of POPC/POPG. The membrane was generated while using the web tool CHARMM-GUI Membrane Builder [31] using a relative composition POPC/POPG of 9 to 1. Each monolayer of the membrane was made of 110 lipids. A periodic simulation cell (X = 85.58 Å, Y = 100.00 Å, Z = 84.44 Å) was built around the entire complex. The charges were assigned at physiological conditions (pH 7.4). The MD simulations were performed using the software YASARA Structure 17.3.30 [32]. We used AMBER14 as a force field with long-ranged PME potential and a cutoff of 8.0 Å. The simulation box was filled with TIP3P water, choosing a density of 0.997 g/mL. The system was neutralized with NaCl at a concentration of 0.9%. The membranes were equilibrated during 200 ps. After equilibration, both peptides ALY and azoALY were embedded in the membrane, placing the transmembrane regions of the peptides perpendicular to the membrane. Short energy minimization was performed in order to optimize the membrane geometry and fill the membrane pores. The simulation was then initiated at 298 K and integration time steps for intramolecular forces every 1.25 fs. The simulation snapshots were saved at regular time intervals of 100 ps. The total simulation time was 50 ns. 

The bilayer thickness is defined as the average distance between the lipid phosphorus atoms of the opposing leaflets. It was calculated averaging the 50 snapshots of the last 5 ns of simulations. In addition, in order to provide an overview of the local membrane deformation along with the simulation, we used MembPlugIn in VMD [33] to interpolate the lipid head’s positions and compute a 2D thickness map. The fluidity was indirectly measured by calculating the deuterium order parameter (SCD) with the VMD MembPlugin software version 1.1.

The mass density profile indicates the atom distribution of the membrane component along the bilayer. This analysis offers useful information regarding the structural changes in membranes [11]. Profiles are determined by dividing the simulation box along the perpendicular to the z-axis (normal to the bilayer) into several thin portions 1 Å thick, and by finding the mass density of the atoms that are positioned in each portion [34].

### 2.3. Synthesis of Azobenzene-Modified Tyr (azoTyr) and the Peptides ALY and azoALY

All of the reagents and solvents were purchased from Sigma–Aldrich (Milan, Italy) and used without further purification. The ^1^H NMR spectra were recorded with a Bruker DRX/400 spectrometer (Bruker, Billerica, MA, USA). Chemical shifts are reported relative to the residual solvent peak (dimethylsulfoxide-d_6_: δ = 2.50 ppm). Mass spectrometry measurements were performed while using a Q-TOF premier instrument (Waters, Milford, MA, USA) that was equipped with an electrospray ion source and hybrid quadrupole-time of flight analyser. The mass spectra were acquired in positive ion mode, in 50% CH_3_CN solution, over the 200–800 *m*/*z* range. Instrument mass calibration was achieved by a separate injection of 1 mM NaI in 50% CH_3_CN. The data were processed using MassLynx software (Waters, Milford, MA, USA).

The azo-amino acid 3-(*p*-tolyldiazenyl)-*N*-Fmoc-L-tyrosine was synthesized according to a diazo coupling reaction, as illustrated in the following Scheme 1: 

0.011 mol of p-toluidine were suspended in a solution containing 19.2 mL of water and 4.8 mL of HCl 37% (*w*/*w*). The solution was cooled at 0–5 °C and a solution of 0.85 g of sodium nitrite (0.012 mol) dissolved in 2.4 mL of water was added dropwise, obtaining a suspension of the diazonium salt (suspension A). Separately, a solution containing 0.018 g of NaOH and 0.068 g of NaHCO_3_ in 20 mL of water with 2.00 g of Fmoc-L-tyrosine (0.00496 mol) was prepared (solution B). Suspension A was added dropwise to solution B, under stirring at 15 °C, adjusting the pH at 9–10, with the addition of NaOH, if necessary. The system was left reacting for 20 min. A reddish precipitate of the azo compound formed. The crude precipitate was filtered, dried under vacuum, and then crystallized from chloroform. The product was recovered as a dark red microcrystalline powder, with a final yield of 85%. ^1^H NMR (DMSO-d_6_): (δ, ppm) = 11.25 (s, OH); 7.87 (d, 2H); 7.80 (d, 2H); 7.63 (dd, 2H); 7.60 (d, 2H); 7.38 (m, 1H); 7.34 (m, 2H); 7.28 (m, 2H); 7.19 (d, 1H); 6.89 (d, 1H); 4.34 (m, 1H); 4.18 (m, 2H); 3.10, 2.98 (m, 2H); and, 2.40 (s, 3H). HRMS (ESI): *m*/*z*: 522.20 [M^+^H^+^].

Peptides ALY (ALYLAIRKR; Mw = 1103.361 g mol^−1^) and azoALY (ALXLAIRKR; Mw = 1221.497 g mol^−1^) were synthesized by Zhejiang Ontores Biotechnologies Co., Ltd (Shanghai, China). Purity for ALY peptide was 96.44% and for azoALY peptide was 96.27%.

A stock solution of ALY was prepared by dissolving a few μg of the peptide in 1 mL of MilliporeQ water. The exact peptide concentration was calculated from the absorbance at 275 nm while using ε_275_ = 1400 M^−1^ cm^−1^ for the single tyrosine chromophore [35,36]. The solution was filtered through a 0.22 µm pore size syringe filter. 

Similarly, a stock solution of azoALY was prepared and the concentration was calculated from the absorbance at 334 nm using ε_334_ = 36805 M^−1^ cm^−1^ for the single modified tyrosine amino acid. The molar extinction coefficient was determined by measuring the UV-Vis spectrum of a solution of azoTyr at a determined concentration. The solution was filtered through a 0.22 µm pore size syringe filter. 

### 2.4. Permeability Tests with POPC/POPG (9:1) Vesicles

#### 2.4.1. Materials

POPC (1-palmitoyl-2-oleoyl-*sn*-glycero-3-phosphocholine) and POPG (1-palmitoyl-2-oleoyl-*sn*-glycero-3-phospho-(1′-*rac*-glycerol) (sodium salt)) (≥99%) were from Sigma–Aldrich Chemie GmbH (Buchs, Switzerland). The membrane indocarbocyanine dye DiI and phosphate-buffered saline (PBS, pH = 7.2) were obtained from Invitrogen Thermo Fisher Scientific (Thermo-Fisher, Waltham, MA, USA)). Calcein was purchased from Fisher Scientific AG (Wohlen, Switzerland). HRPC (Horseradish peroxidase isoenzyme C, EC 1.11.1.7, RZ A_403_/A_280_ > 3.1) was from Toyobo Enzymes (Osaka, Japan). The diammonium salt of ABTS^2−^ (2,2′-azinobis(3-ethylbenzothiazoline-6-sulfonate)) and Triton X-100 were purchased from Sigma–Aldrich Chemie GmbH (Buchs, Switzerland). Hydrogen peroxide (H_2_O_2_, 30%) was from Acros Organics Fisher Scientific AG (Wohlen, Switzerland), 4-morpholineethanesulfonic acid (MES ≥ 99%) was purchased from Fluka Fisher Scientific AG (Reinach, Switzerland). Sepharose 4B and chloroform (stabilized with ethanol, 99.8%) were purchased from Sigma–Aldrich Chemie GmbH (Buchs, Switzerland).

#### 2.4.2. Calcein Release Test with Giant Unilamellar Vesicles (GUVs)

Wide-field microscopy was performed with an inverted microscope (IX70, Olympus America, Melville, NY, USA) equipped with a mercury lamp and a 40×/0.65 NA air objective lens. The images were recorded with an EMCCD camera (iXon DV887, Andor Technology, South Windsor, CT)). The fluorescence intensity of the calcein within three separate vesicles was monitored while using a confocal laser-scanning microscope (Axiovert 200 M, Zeiss, Oberkochen, Germany), and an appropriate optical filter sets for DiI and calcein. The GUVs were prepared by electroformation following the procedure that was originally described by Angelova et al. [37], but conducted in a modified chamber [38]. Briefly, the setup used for the preparation of the vesicles consisted of two conductive indium tin oxide (ITO) coated glasses separated by a 1.5 mm thick silicone rubber spacer to maximize the yield of vesicles. POPC/POPG lipids (9:1 mol/mol) were dissolved in chloroform/methanol (9:1 *v*/*v*) at a concentration of 1 mM. The orange-red fluorescent dye DiI was added at a concentration of 1 μM. A drop of 2.5 μL of the mixture was deposited on one of the conductively coated glasses, and this was repeated in 12 locations. The lipid film was then dried under vacuum overnight and hydrated with MilliQ water containing 10 μM calcein. The chambers were sealed by a second ITO slide and held at 60 °C within a custom-built heating device. GUVs were formed by applying 0.7 V at a frequency of 10 Hz for 4 h while using a function generator. After applying 1 V at 4 Hz for 30 min. to detach the vesicles from the surface, harvesting was achieved by careful pipetting. GUVs were stored at room temperature and used within 48 h. For the analysis of calcein release from the GUVs, we used a microfluidic platform that was able to trap single GUVs in an array of chambers [39] (see Appendix A). By exchanging the solution inside the chip and subsequently opening the valves, it is possible to perform fast kinetic studies from the seconds to minutes timescale. GUVs containing fluorescent calcein were trapped in order to investigate the membrane perturbation and subsequent leakage of calcein across the membrane. Because of the hydrophilic nature of calcein, it does not permeate the membrane and remains encapsulated. Once trapped, the GUVs remain stable for long times (at least 12 h), and no significant deformation or rupture of the vesicles was observed. The solution was then exchanged with MilliQ water while using a syringe-pump (neMESYS, Cetoni, Germany) until only calcein fluorescence inside GUV was visible and no more calcein was detected in the surrounding solution (See Appendix A). The same pump was used to exchange the aqueous solution outside the donuts with an aqueous solution of the peptide, with a total flow rate of 5 μL min^−1^. The release measured in the absence of the peptide was used as the control, and the amount of calcein within the GUV remained constant, indicating that photobleaching did not occur. The error bars were calculated from the standard error, estimated by population standard deviation divided by the square root of the sample number. The number of individual experiments at a given peptide concentration was n = 3.

#### 2.4.3. Enzymatic Permeability Assay with Large Unilamellar Vesicles (LUVs)

LUVs that were composed of POPC/POPG (9:1 mol/mol) and loaded with HRPC were prepared by lipid film hydration, followed by mechanical extrusion using for final extrusions 200 nm polycarbonate membranes, in a similar way described before for POPC vesicles containing HRPC [40]. Weighted amounts of lipids (20 mM) were first dissolved in chloroform that was stabilized with ethanol and then mixed in a 100 mL round bottom flask. The solvent was removed by rotatory evaporation at 35 °C and dried in high vacuum overnight. The obtained thin lipid film was hydrated with 3.5 mL of 20 μM HRPC solution in 10 mM (MES) buffer (pH = 5.0) at room temperature. The hydration was made by gentle agitation of the flask to exclude enzyme denaturation. Ten freezing-thawing cycles were carried out in order to homogenize and equilibrate the enzyme-containing multilamellar vesicles suspension (MLV), cooling the flask in liquid nitrogen for 20 seconds followed by placing the flask in a warm (50 °C) water bath for 30 seconds. The MLV suspension was first extruded 10 times through track-etch polycarbonate membranes with cylindrical pores of 400 nm size under moderate pressure (three-bar) at room temperature, followed by extrusion with a 200 nm membrane increasing the pressure to five-bar at the same temperature conditions. The non-entrapped enzyme molecules were separated from the enzyme-containing vesicles by size-exclusion chromatography by using a 2 × 20 cm glass column that was filled with Sepharose 4B equilibrated with MES buffer (pH = 5), the flow rate of 0.5 mL/min. 

We performed turbidity measurements of each eluted fraction to select the most concentrated fraction of HRPC-containing vesicles with the lowest amount of free enzyme (pI (HRPC) ≈ 10) possibly bound to the outer membrane of the anionic vesicles (see Appendix A for details). The fraction with the highest optical density at 403 nm was used in order to establish the release of HRPC from enzyme containing POPC/POPG vesicles.

The activity of HRPC was measured with ABTS^2−^. ABTS^2−^ is frequently used as a sensitive chromogenic substrate for the quantification of horseradish peroxidase [40,41]. ABTS^2−^ has an absorption maximum at 340 nm. Depending on the experimental conditions, HRPC catalyses the oxidation of ABTS^2−^ forming a stable nitrogen-centered radical cation, so that the obtained product is overall negatively charged, ABTS^•−^. ABTS^•−^ has an absorption maximum at 414 nm and three other bands centered around 650, 735, and 814 nm. Therefore, UV/Vis-spectroscopy is most convenient for determining the catalytic activity of HRPC by following the rate of reaction product formation [42,43]. The rate of ABTS^•−^ formation (monitored at λ_max_ = 414 nm) during the first 5 min. of the reaction was followed spectrophotometrically and found to linearly depend on the HRPC concentration between 50 and 350 pM under the conditions used (see the curve in Appendix A). The following reaction conditions for the spectrophotometric quantification of the HRPC activity were found to be appropriate: [ABTS^2−^]_0_ = 0.25 mM, [peptide] = 4 μM, [H_2_O_2_]_0_ = 80 μM, 100 μL of pooled fraction diluted at 25 °C in MES buffer pH 5, for a total assay volume of 1 mL and reaction time of 5 min. The activity measurements were carried out, as follows: buffer to reach the assay volume of 1 mL, ABTS^2−^ (final concentration of 0.25 mM), and HRPC stock solution was mixed in a reaction tube. Immediately before the spectrophotometric analysis, H_2_O_2_ was added (final concentration of 80 μM). The mixture was transferred into a polystyrene cuvette (path length = 1 cm) and the development of the absorption spectrum of the reaction mixture was monitored as a function of time (see Appendix A) while using a diode array spectrophotometer (Specord 5600 Analytik Jena, Jena Germany). The spectra were recorded every 10 sec immediately after the initiation of the reaction (up to 5 min.). All of the measurements were carried out three times. Triton X-100 was used as the control. For the control assay, we used [ABTS^2−^]_0_ = 0.25 mM, [H_2_O_2_]_0_ = 80 μM, 100 μL of pooled fraction 100× diluted and 20 μL 5% *v*/*v* Triton X-100 at 25 °C in MES buffer, pH 5. Because of the delay of inactivation in the course of substrate reaction [44], we built a calibration curve in the presence of Triton X-100 (see Appendix A). For the assay, we used a surfactant concentration of 0.1% in the assay mixture containing HRPC and ABTS^2−^ solutions. After 5 min. from Triton X-100 addition, the reaction was started adding hydrogen peroxide, and the enzymatic reaction was followed by UV/Vis spectrophotometry.

## 3. Results and Discussion

### 3.1. Peptide Localization within Lipid Bilayers: a MD Simulation Analysis

In this study, we analysed two peptides, a natural peptide, named ALY, and a modified peptide, azoALY. They both have a short sequence of nine amino acids that fold into an alpha helix structure. The azo amino acid slightly deforms the helix structure adopted by azoALY. In the MD simulations, the hydrophobic regions of the peptides ALY and azoALY were embedded perpendicularly to the membrane bilayer of POPC/POPG (9:1) at the starting point. We used an online tool to predict the position of membrane proteins (https://opm.phar.umich.edu). The model suggested the insertion for both peptides in membrane with a tilt angle of 48° and with the hydrophobic portion (Ala-Leu-Tyr-Leu-Ala) immersed in the core membrane. The charged portion of the peptide (Arg-Lys-Arg) is external to the membrane for both peptides (see Appendix A). After the first 200 ps of the simulation, the ALY peptide comes out of the bilayer and it is located between the head groups and the water phase in the outer part of the membrane. In contrast, the azoALY peptide is anchored to the membrane for the whole simulation time (see Appendix A). The density profile indicates the position of the different atoms of the system along the normal phospholipid bilayer in the last 5 ns of a simulation of 50 ns. In Figure 1 and Figure 2, the density profiles of ALY and azoALY (normalized to 1 by water mass) in the POPC/POPG membrane is reported. The cyan line represents the water continuum; it is outside the bilayer, ensuring membrane hardness. The phosphorus and nitrogen atoms (continuous blue and dashed blue line. respectively) represent the head group of the phospholipids; the dashed black line indicates the carbon chains of the phospholipid, while the black line represents the terminal carbons. After a simulation time of 45 ns, the ALY peptide (red line in Figure 1) is placed below the phosphorus atoms between the head group and the water phase, as seen in Figure 1. In contrast, the peptide azoALY (red line in Figure 2) is anchored to the core membrane for the same simulation time (from 45 to 50 ns).

For both peptides, the membrane thickness of the peptide/membrane systems was measured in order to estimate the perturbation of the membrane packing induced by the peptide, when only considering the lipids surrounding the peptide with a distance of 4 Å. The membrane thickness was measured as the average distance between the center of mass of the phosphate atoms of the inner layer and the outer layer. The thickness of the pure POPC/POPG bilayer did not change during the 50 ns simulation, suggesting that the system was equilibrated with an initial value of 38.5 ± 0.3 Å. In the presence of the two peptides, in the last 5 ns of the simulations, the membranes thickness values were 38.7 ± 0.3 Å with ALY and 36.4 ± 0.5 with azoALY. The modified peptide causes a reduction of the membrane thickness. This different perturbation is due to different peptide-membrane interactions. The peptide ALY is placed between the water molecules and head groups, and it does not cause perturbation of the membrane packing. On the other hand, azoALY is anchored to the membrane core, which affects the invagination of the lipid surrounding and produces a significant reduction of the membrane thickness.

From Figure 2, in fact, the peptide azoALY (red line) is in the same position as the terminal carbon atoms of the lipid tails. This simulation suggests that the peptide is anchored to the core of the membrane in the last 5 ns of the simulation time. The azo group has a significant influence on the peptide-membrane interaction: due to its rigid and planar structure, the azo amino acid permeates the membrane, causing disorder in the lipid leaflet. We performed a deuterium order parameter (SCD) analysis. SCD studies provide comprehensive information on membrane fluidity of lipophilic phospholipid chains near peptide molecules to assess whether the modified peptide causes a change in membrane fluidity. The analysis, performed on the surrounding lipids with a distance of 5 Å from the peptides, confirmed a different behavior of the two peptides. azoALY increases the mobility in the lipid head more than peptide ALY. The results are shown in Appendix A section.

The thickness maps in Figure 3 show the local membrane deformation along with the simulation. The ALY peptide has a minimal effect on membrane thickness with a constant fluctuation of lipids around a value of 38 Å (Figure 3a). The azo-modified peptide causes the invagination of the membrane, reducing the thickness of the surrounding lipids that are represented by the blue hole in top leaflet of the membrane (Figure 3b).

### 3.2. Experimental Membrane Permeability Measurements

#### 3.2.1. Calcein Leakage from GUVs

The membrane perturbing effects of the modified peptide azoALY were also investigated while using calcein leakage experiments in a suspension of giant unilamellar vesicles (GUVs) formed by POPC/POPG (9:1, mol/mol). In this assay, a fluorescent probe (calcein) was employed, and its release from the GUVs was followed by measuring its fluorescence intensity inside the vesicles. Excessive leakage upon adding peptides or other compounds to the GUVs indicates instability in the structure of the vesicle membrane (alteration of the packing of the lipids). This, in turn, suggests that the peptide added to the GUVs strongly interacts with the lipid membrane. GUVs represent an excellent model representing the lipid membrane of biological cell membranes and they have become an essential tool in biophysical research [45,46]. The experiments were carried out using two different concentrations of peptides: 1 μM and 50 μM of ALY and azoALY. Figure 4 shows a representative confocal image series after the addition of 50 μM azoALY, where calcein diffuses out of the GUVs via a decrease in fluorescence intensity.

After 15 min. of incubation with the peptides, the GUVs preserved their membrane structure (i.e., no visible defects), so we exclude the destruction of the membrane or micron-sized pore formation. However, a considerable amount of calcein remained entrapped inside the vesicles after 15 min. under the used conditions. The analysed GUVs were selected to have the same diameter of approximately 12 μm to maintain the same surface area for comparable results. The mean pixel intensities within each GUV were normalized and plotted against time (Figure 5).

At low concentrations of ALY peptide (1 μM), during the first 15 min. of exposure, the calcein fluorescence undergoes a decrease of 17%. Nevertheless, while using the same concentration of the modified azoALY peptide, a decrease of 31% in intensity was observed at the same time. In contrast, when high concentrations of peptides were used (50 μM), it was possible to observe that the percentage of calcein release is similar for both of the peptides. In that case, the addition of azoALY resulted in a faster calcein release than the reference peptide ALY. In fact, during the first three minutes of the experiment, 25% of calcein leakage was observed, though, after 15 min., both of the peptides reached the same calcein leakage.

#### 3.2.2. Permeabilization of LUVs

We performed an enzymatic assay for detecting possible changes in the permeability of LUVs to further evaluate the membrane permeability increase upon peptide addition to the calcein-containing GUVs. For this purpose, LUVs containing entrapped horseradish peroxidase isoenzyme C (HRPC) were prepared. Afterward, the chromogenic substrate ABTS^2−^ and hydrogen peroxide (H_2_O_2_) were added to the enzyme-containing vesicles. An unperturbed phospholipid membrane is impermeable for HRPC and ABTS^2−^. The HRPC molecules cannot cross the membrane due to their large size, and ABTS^2−^ molecules cannot move from the external bulk solution into the interior of the vesicles due to their negative charge. On the contrary, H_2_O_2_, being small and uncharged, can easily permeate across fluid phospholipid bilayers [47]. The release of HRPC from the LUVs and/or the uptake of ABTS^2−^ by the LUVs after peptide addition can be conveniently monitored by UV-vis spectrophotometry as the HRPC-catalysed oxidation of ABTS^2−^ to ABTS^•−^ results in increased absorbance (see Figure 6 for a schematic representation of the permeability assay). 

HRPC was entrapped in LUVs with a diameter of about 200 nm. We used the same formulation for the vesicles, as in the case of the experiments with GUVs, i.e., POPC/POPG (9:1). The amount of entrapped HRPC was quantified by measuring the HRPC activity after adding Triton X-100 (0.1 vol%) that causes immediate vesicles destruction and the release of the entrapped enzyme into the assay solution (dilution factor 2000×). The increase of membrane permeability (without added Triton X-100, but with added peptides) was estimated by measuring the enzyme activity with ABTS^2−^ and H_2_O_2_ as HRPC substrates, monitoring the formation of ABTS^•−^. 

We analysed the interactions of ALY and azoALY with POPC/POPG (9:1) LUVs containing entrapped HRPC (dilution factor 100×). We used as blank the system POPC/POPG (9:1) LUVs with entrapped HRPC, without peptides (dilution factor 100×), in order to exclude the presence of enzyme molecules bound to the outer membrane surface of the vesicles. Measurements were performed after different incubation times (0, 5, 10 min.) to estimate a possible time dependence of the interactions and understand if and how much the azo-amino acid inserted in the amino acidic sequence could affect the membrane permeability. The preparation of the vesicles with entrapped enzyme was repeated three times and, for each mixture, we evaluated the membrane perturbation due to the action of ALY and azoALY. In all cases, we obtained analogous enzyme leakage. Therefore, we excluded the possibility of false-positive results by the oxidation of the phenol moiety (in tyrosine) and azo-group in the peptides. Moreover, we excluded a possible shift of the characteristic peaks of the oxidized ABTS^2−^ form (414 nm, 650, 735, and 814 nm) due to the presence of peptides. We also checked if a mixture of “empty” vesicles and peptide influences the enzymatic reaction and we exclude enzyme inhibition by the assay mixture. 

Linear regression of ABTS^•−^ absorbance at λ_max_ = 414 nm as a function of reaction time allowed for the determination of the HRPC activity, read as the slope of the linear fit (dA_414nm_/dt), see Figure 7.

The enzymatic activity for the blank was very low, and the corresponding HRPC concentration in the assay solution was lower than 50 pM (for the calibration curve see Appendix A). Besides, the activity for this system was not time-dependent: the slope value did not change for the three different incubation times. This means that the amount of HRPC that was possibly bound to the outer membrane was quite low; additionally, there was no leakage of the enzyme or uptake of the substrate. *Vice versa*, a significant increase in enzyme activity was observed after 5 min. of incubation in enzyme-containing vesicles plus the surfactant Triton X-100 (0.1%). Analysing the calibration curve in the presence of Triton X-100 (see Appendix A), the concentration of the entrapped enzyme was estimated to be around 400 nM (when considering a dilution factor 2000×). The ALY peptide produced a small increase in enzyme activity. At the beginning (incubation 0 min.), the effect on the membrane permeability was comparable to the blank. After 10 minutes of incubation, we recorded an enzymatic activity corresponding to an enzyme concentration lower than 5 nM (dilution factor 100×). The situation was completely different while using the modified peptide azoALY: an immediate significant increase in HRPC activity was registered, and the activity increase was time dependent. When considering a dilution factor 100×, the amount of enzyme released for 0 min. of incubation corresponds to an enzyme concentration of HRPC around 10 nM, for 5 min. was 25 nM and for 10 min. was 33 nM. We could estimate an increase of enzyme release or ABTS^2−^ uptake time dependent with an enzyme concentration in the pooled fraction of 10 nM, 25 nM, and 32.5 nM, respectively.

The HRPC/LUV experiments indicate a time-dependent increased membrane permeability upon the addition of azoALY. However, it remains to be clarified whether the higher enzymatic activity is ascribed to ABTS^2−^ uptake or HRPC release from the vesicles.

## 4. Conclusions

The combined use of MD simulations, together with experimental tests, allowed for us to investigate the interaction of a natural peptide and its azobenzene modified analog with artificial membrane models. We have designed a novel peptide (azoALY) by modifying a natural membrane-active peptide (ALY) with a tyrosine containing an azo-group in the side chain. Computational and experimental permeability studies with model membrane systems (vesicles of POPC and POPG at a molar ration of 9:1) suggest that the insertion of the modified tyrosine residue in the peptide sequence increases the peptide affinity to the vesicle membrane. A different membrane permeability for ALY and azoALY was observed in experimental membrane permeability assays while using GUVs and LUVs: increased calcein release form GUVs and increased HRPC leakage from the LUVs or increased ABTS^2−^ uptake by the LUVs. The MD simulations on model membranes support these results. The all-atom molecular dynamics showed a higher membrane perturbation by the modified peptide (azoALY) than the unmodified peptide (ALY) under the same conditions. The azoALY peptide penetrates the hydrophobic core of the bilayers, while the ALY structure only interacts with the lipid headgroups, causing a lower extent of membrane alteration. As a more general consideration, our investigations showed that the results obtained from physical-chemical in vitro investigations with simple membrane model systems might not directly correlate with the in vivo behavior of biological membranes of living cells. The vesicle and the bacterial membrane compositions were very different, and the actual concentrations in terms of peptides and membrane components were difficult to compare. Similar conclusions were drawn before from other studies [48,49].

Still, based on the obtained results, the presence of a modified amino acid residue appears to be essential in determining the perturbation effect of the azobenzene peptide membrane. Under such conditions, modified peptides could increase peptide-membrane interaction and complete their biological task (for example, antimicrobial agents). The novel semi-synthetic peptide could act as a prodrug, generating in vivo the antimicrobial azo compound. The preliminary results of membrane permeability underlined a higher membrane perturbation with time-dependence leakage from lipid vesicles after interaction with modified peptide as compared to the unmodified peptide under the same conditions. In conclusion, we evidenced a different behavior of the natural and the azo-modified peptide, and we suggested the use of azo-modified amino acid moieties to increase the membrane modification. Although only one peptide pair and one membrane model were studied, the different effects on membrane perturbation suggest that these effects are potentially of broader significance. Additional work is needed in order to clarify the possible generalization of the results that were observed in this study.

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
