# Peer review of "Study of the Interaction of a Novel Semi-Synthetic Peptide with Model Lipid Membranes"

_membranes, 2020, doi:10.3390/membranes10100294_

Round 1

Reviewer 1 Report

The authors have presented combined computational and experimental data to show that different peptides can differentially perturb membranes and change membrane permeability. The manuscript in its current form, has major weaknesses that need to be addressed before it can be considered for publication:

  1. The peptide construct azoALY is artificial, and the connection to real biological systems needs to be shown for these results to be relevant. If the authors do not have data connecting these results to real biological systems, at a minimum the authors need to discuss that the impact on membrane structure can be very different for real proteins vs these linear peptide constructs with references to data in the literature, e.g., in Biophys J 2011, "Quantitative Modeling of Membrane Deformations by Multi-Helical Membrane Proteins".
  2. The simple point that peptides can affect membrane structure is already known for biologically relevant peptides which need to be referenced, e.g., see the work with Dynorphin in Khelashvilli et al J Phys Chem B 2008, "Cholesterol Modulates the Membrane Effects and Spatial Organization of Membrane-Penetrating Ligands for G-Protein Coupled Receptors". The key novelty in this paper, so far as I can see, is that the change in membrane structure can change membrane permeability. 
  3. The analysis of the MD simulations is inadequate. Averaging over the last 5 ns of the simulation is fine, but to do so with only 5 snapshots total makes for inadequate sampling of the trajectory. Given the fluctuations in lipid membranes, 5 ns needs to be sampled at a much higher rate for robust results, I would suggest 50 frames for the 5 ns. Moreover, showing the membrane thickness in Fig. 1 is a good start, but the authors need to minimally show that the perturbation in thickness vs. distance from the peptide, the order parameters, as well as the impact on membrane structures with representative or final snapshots, e.g., see the two papers mentioned above.

Overall, the side-by-side use of MD simulations and experimental data is a key strength of this paper, and the paper has the potential to be quite impactful -- but can be considered for publication only after substantially improving the analysis of the MD simulations, the connection to existing literature, and connection to real biological systems.

Author Response

First of all, we express our gratitude to the anonymous reviewer that used her/his valid time to improve the quality of the manuscript. In the following, we detail our answers:

The authors have presented combined computational and experimental data to show that different peptides can differentially perturb membranes and change membrane permeability. The manuscript in its current form, has major weaknesses that need to be addressed before it can be considered for publication:

1.The peptide construct azoALY is artificial, and the connection to real biological systems needs to be shown for these results to be relevant. If the authors do not have data connecting these results to real biological systems, at a minimum the authors need to discuss that the impact on membrane structure can be very different for real proteins vs these linear peptide constructs with references to data in the literature, e.g., in Biophys J 2011, "Quantitative Modeling of Membrane Deformations by Multi-Helical Membrane Proteins".

We agree with the reviewer. Indeed, for space reasons we had limited the number of biophysical in silico characterizations of membranes. Based on the scientific literature, like the example suggested by the Reviewer, we inserted a more precise characterization of the structural changes of the membrane surface under the action of the two peptides (see lines 55-57, lines 326-342, new figure 3, and figure S7 in Supplementary Materials.)

2. The simple point that peptides can affect membrane structure is already known for biologically relevant peptides which need to be referenced, e.g., see the work with Dynorphin in Khelashvilli et al J Phys Chem B 2008, "Cholesterol Modulates the Membrane Effects and Spatial Organization of Membrane-Penetrating Ligands for G-Protein Coupled Receptors". The key novelty in this paper, so far as I can see, is that the change in membrane structure can change membrane permeability. 

It is known that linear peptides affect membrane structure. Little is known about these new constructs in which tyrosine has been modified with azobenzene.

The objective of this work was to determine how the presence of this group can affect the membrane structure and its physical state. We have added some references accordingly.

3.The analysis of the MD simulations is inadequate. Averaging over the last 5 ns of the simulation is fine, but to do so with only 5 snapshots total makes for inadequate sampling of the trajectory. Given the fluctuations in lipid membranes, 5 ns needs to be sampled at a much higher rate for robust results, I would suggest 50 frames for the 5 ns. Moreover, showing the membrane thickness in Fig. 1 is a good start, but the authors need to minimally show that the perturbation in thickness vs. distance from the peptide, the order parameters, as well as the impact on membrane structures with representative or final snapshots, e.g., see the two papers mentioned above.

Overall, the side-by-side use of MD simulations and experimental data is a key strength of this paper, and the paper has the potential to be quite impactful -- but can be considered for publication only after substantially improving the analysis of the MD simulations, the connection to existing literature, and connection to real biological systems.

According to the Reviewer’s suggestion, we have recalculated the membrane thickness using the last 50 frames of the MD (lines 128-133). The results remained substantially in line with what was already shown. We changed the text with the new values (line 306).

To establish the perturbation of the membrane lipids in relation to the distance from the peptide, we have also calculated the local thickness map (results are shown in new figure 3).

To show the changes of the membrane peptide position, we have added for both peptides an image with the first and last snapshot of the dynamics (see new figure S6 in Supp. Mat.).

The fluidity was measured indirectly by calculating the deuterium order parameter (SCD) with the VMD MembPlugin software (see new figure S7 in Supp. Mat.).

We have adapted the text accordingly and extended the referenced literature. 

Reviewer 2 Report

The paper deals with the interaction of a linear nonapeptide and its chemically modified azo-version with model lipid membranes. The subject is the comparison of the membrane penetration properties of the two peptides experimentally using lipid vesicles and performing molecular dynamic simulations. The experiments involving dye leakage from GUV and enzymatic assay in the case of LUV are carried out properly. Although, the explanation considering the molecular structure is missing. The work and its presentation is correct but in my opinion the contribution to the understanding of mechanism of membrane peptide interaction is not substantial.

Some questions and comments:

  1. 2 on the selection of POPC/POPG lipids
    As it is mentioned POPC component is not characteristic of bacterial membrane. Its application is rather explained by the facile formation of vesicles due to its low phase transition temperature.
  2. 7-8 MD simulation
    The equilibrium location of peptide in the lipid bilayer determined would be more convincing if the start position is not in the inner region of the bilayer but the surrounding water providing similar condition to the experimental permeation test.
    From the course performed here it can hardly be concluded that “azo amino acid permeates the membrane” or “penetrates the hydrophobic core of the bilayer”. Both peptides have a highly charged polar head (RKR). Where is this in the case of azoALY?

The change in the bilayer thickness caused by interaction with azoALY is considered as strong membrane perturbation however that is not more than 6%. At the same time the concentration profiles of various atoms representing the structure of the bilayer and arrangement of lipid molecules are highly similar for the two peptides in Figure 2. So I can not find supported the statement on the enhanced perturbation of lipid bilayer and “disorder in the lipid leaflet” based of MD simulations.

  1. 10 Fig. 4
    There is a significant difference between the calcein release caused by the two peptides at one studied concentration, and not in general. So the sentence related to this finding in the Abstract is misleading.
  2. 11 Fig. 12
    How is explained the lower enzyme activity for the case of surfactant (Triton x-100) application than azoALY?

Author Response

First of all, we express our gratitude to the anonymous reviewer that used her/his valid time to improve the quality of the manuscript. In the following we detail our answers

The paper deals with the interaction of a linear nonapeptide and its chemically modified azo-version with model lipid membranes. The subject is the comparison of the membrane penetration properties of the two peptides experimentally using lipid vesicles and performing molecular dynamic simulations. The experiments involving dye leakage from GUV and enzymatic assay in the case of LUV are carried out properly. Although, the explanation considering the molecular structure is missing. The work and its presentation is correct but in my opinion the contribution to the understanding of mechanism of membrane peptide interaction is not substantial.

Some questions and comments:

1. 2 on the selection of POPC/POPG lipids
As it is mentioned POPC component is not characteristic of bacterial membrane. Its application is rather explained by the facile formation of vesicles due to its low phase transition temperature.

We have rephrased the text accordingly in lines 81-82.

2. 7-8 MD simulation
The equilibrium location of peptide in the lipid bilayer determined would be more convincing if the start position is not in the inner region of the bilayer but the surrounding water providing similar condition to the experimental permeation test.
From the course performed here it can hardly be concluded that “azo amino acid permeates the membrane” or “penetrates the hydrophobic core of the bilayer”. Both peptides have a highly charged polar head (RKR). Where is this in the case of azoALY?

We agree with the reviewer that the initial conditions of molecular dynamics do not correspond to those of the calcein release tests. However, the time needed to reach equilibrium (in the micro/millisecond range) can be very shortened by choosing an initial configuration corresponding to the lower transfer free energy of the peptide from water to the lipid bilayer.

For both peptides, we used the web tool (https://opm.phar.umich.edu) for evaluating the position and angle provided by the model. To reduce the number of mismatches with the membrane lipids we placed the peptides normally at the membrane with the minimum distance from the OPM.

We have added some sentences to better explain the choice (lines 281-285).

The change in the bilayer thickness caused by interaction with azoALY is considered as strong membrane perturbation however that is not more than 6%. At the same time the concentration profiles of various atoms representing the structure of the bilayer and arrangement of lipid molecules are highly similar for the two peptides in Figure 2. So I can not find supported the statement on the enhanced perturbation of lipid bilayer and “disorder in the lipid leaflet” based of MD simulations.

1. 10 Fig. 4
There is a significant difference between the calcein release caused by the two peptides at one studied concentration, and not in general. So the sentence related to this finding in the Abstract is misleading.

We have rephrased the text accordingly in the abstract (line 28). Moreover, we inserted a more precise characterization of the structural changes of the membrane surface under the action of the two peptides (see lines 326-342, new figure 3, and figures S6 and S7 in Supplementary Materials.)

2. 11 Fig. 12
How is explained the lower enzyme activity for the case of surfactant (Triton x-100) application than azoALY?

The surfactant showed higher enzyme activity considering the dilution factor of 2000 x.

We have adjusted the caption of new figure 7 to better highlight this point.

Reviewer 3 Report

This is an interesting article showing the investigation of the interaction of a natural peptide and its azobenzene modified analog with artificial membrane models. The Authors show that results obtained from physical-chemical in vitro investigations with simple membrane model systems might not directly correlate with the in vivo behavior of biological membranes of living cells. 

The idea is interesting for future novel semi-synthetic peptide research and deserves more exploratory interventions.

Author Response

We express our gratitude to the anonymous reviewer that used her/his valid time to carefully read the manuscript and for her/his positive judgment.

Round 2

Reviewer 1 Report

Thanks to the authors for revising the manuscript. The analysis of the MD simulations is now richer and sufficiently robust, I recommend that the manuscript be accepted for publication.